# Lipidomic Signature of Healthy Diet Adherence and Its Association with Cardiometabolic Risk in American Adults

**DOI:** 10.3390/nu16233995

**Published:** 2024-11-22

**Authors:** Loni Berkowitz, Guadalupe Echeverría, Cristian Salazar, Cristian Faúndez, Christopher L. Coe, Carol Ryff, Attilio Rigotti

**Affiliations:** 1Department of Nutrition, Diabetes and Metabolism, School of Medicine, Pontificia Universidad Católica de Chile, Santiago 8330077, Chile; 2Department of Psychology, University of Wisconsin-Madison, Madison, WI 53706, USA; ccoe@wisc.edu (C.L.C.);; 3Institute on Aging, University of Wisconsin-Madison, Madison, WI 53706, USA

**Keywords:** healthy diet, cardiovascular risk, lipidomics, lipid signature, polyunsaturated fatty acids

## Abstract

Background: The aim of this study was to identify the blood lipidomic profile associated with a healthy eating pattern in a middle-aged US population sample and to determine its relationship with metabolic disorders and cardiovascular risk (CVR). Methodology: Self-reported information about diet and blood samples were obtained from 2114 adult participants in the Midlife in the United States study (MIDUS). Food intake data were used to design a Healthy Diet Index (MIDUS-HEI) and to evaluate the predictive value by examining its association with health variables. The associated lipid signature (HEI-LS) was constructed using Lasso regression, from lipidomic data (LC/MS). Associations between HEI-LS, cardiometabolic biomarkers, and estimated CVR were assessed using multiple linear regression. Results: MIDUS-HEI score was a robust indicator of dietary quality and inversely associated with body mass index (*p* < 0.001) and metabolic syndrome (*p* = 0.012). A lipidomic signature comprising 57 distinct lipid species was highly correlated with the MIDUS-HEI score (r = 0.39, *p* < 10⁻^16^). It was characterized by lower levels of saturated fatty acid and adrenic acid (*n*-6) and higher levels of docosahexaenoic acid (*n*-3). Healthier HEI-LS scores were strongly associated with better cardiometabolic indicators and lower estimated CVR (OR 0.89 CI 95% 0.87–0.91). Conclusions: The MIDUS-HEI effectively assessed dietary quality, confirming the link between poor diet quality and metabolic disorders in American population. Lipidomic profiling offered an objective assessment of dietary patterns and provided insights into the relationship between diet quality, metabolic responses, and CVR. This approach supports precision nutrition strategies for at-risk populations.

## 1. Introduction

Cardiometabolic diseases, including cardiovascular diseases and diabetes, have reached an alarming prevalence worldwide, significantly contributing to global morbidity and mortality [1]. Among the modifiable lifestyle risk factors, diet emerges as the principal determinant of cardiometabolic health [2]. Numerous studies have demonstrated that adherence to healthy dietary patterns, such as the Mediterranean diet and the Dietary Approaches to Stop Hypertension diet (DASH), can substantially reduce cardiovascular risk [3].

Evaluating adherence to a healthy diet is crucial for understanding its impact on health outcomes. An individual’s diet is made up of a complex mix of different foods, which interact to determine their long-term effect on health and disease risk. Indeed, the overall quality of the diet influences the risk of chronic diseases to a greater extent than specific macronutrients [3,4,5]. Rather than focusing on individual nutrients, assessing dietary patterns provides a more representative picture of overall diet quality. Several indices have been developed to quantify dietary patterns, including the Healthy Eating Index (HEI), the Alternate Healthy Eating Index (AHEI), the Mediterranean Diet Score (MDS), and the DASH diet score. These indices provide structured approaches to assess dietary quality and adherence to recommended dietary guidelines, thus enabling evaluation of their relationship with health outcomes. For example, AHEI has been inversely related to coronary heart disease, type 2 diabetes, cancer, chronic obstructive pulmonary disease, and depression [6,7,8]. Similarly, Mediterranean diet adherence scores have been associated with lower incidence of obesity, metabolic syndrome (MetS), diabetes, cancer, and cardiovascular and neurodegenerative diseases, as well as overall mortality [9].

However, traditional methods for measuring dietary intake, such as food frequency questionnaires (including dietary pattern scores), 24 h dietary recalls, and dietary records, have several limitations. These approaches include reliance on self-reported data, which can introduce recall bias and inaccuracies, and difficulty in capturing the complexity of dietary variation over time and the actual metabolic interaction with an individual’s physiology [10].

Recent advances in metabolomic and lipidomic assay platforms, which allow for the comprehensive measurement of a large number of metabolites and lipids in a sample, offer powerful tools for objectively assessing adherence to dietary patterns. By analyzing metabolomic changes, researchers can indeed identify specific metabolites that reflect dietary intake and adherence [11]. This approach enhances the accuracy of dietary assessments while also providing insight into the potential interactions between dietary constituents, metabolism, and health. For example, a Spanish research team identified a metabolic signature comprising 67 metabolites that sensitively reflected adherence to a Mediterranean diet and predicted the reduction in risk for cardiovascular disease (CVD) [12].

Among the myriad metabolites present in the human body, lipids play a key role in the development and progression of unhealthy diet-related diseases. However, the nature of their involvement is not fully understood as we have come to learn more about the thousands of lipid species that comprise the structure of cells and traffic in circulation [13]. Despite prior advances, there is need to deepen understanding of how specific lipid species are influenced by diet and their involvement in sustaining health and pathophysiology. Characterizing lipid species associated with a healthier diet can elucidate their contributions to cardiometabolic resilience and inform about targeted dietary interventions.

The aim of this study was to identify the blood lipidomic profile associated with a healthy eating pattern in a middle-aged US population sample and to determine its relationship with metabolic disorders and cardiovascular risk (CVR).

## 2. Materials and Methods

### 2.1. Study Sample

The MIDUS project, initiated in 1995, has been investigating the relationships between behavioral factors, psychosocial factors, and health in a representative sample of Americans aged 25 to 74 years for several decades (MIDUS Core). During the second wave, initiated in 2004, the MIDUS Core sample was expanded to include more African Americans, and biomarkers were collected from a subsample of participants (*n* = 1255). In 2012, a new national sample (MIDUS Refresher) was recruited, with biomarkers collected from a subsample (*n* = 863). In both groups, participants completed overnight clinic-based biomarker data collection at one of three regional sites (Los Angeles, CA, USA; Madison, WI, USA; and Washington, DC, USA). Clinical assessments were conducted using the same methods for both cohorts at all locations, and biomarker assays were all run at the same testing laboratories. Individuals who did not provide the information needed to calculate the healthy eating index were excluded. Thus, the final analysis was based on a sample of a total of 2114 participants: 1252 from the MIDUS Core and 862 from the MIDUS Refresher phase.

### 2.2. Dietary Assessment

Food intake data were obtained via medical history interview conducted by project staff during the biomarker clinic visit. These computerized questions were part of the Current Health Practices—Diet and Exercise section of the interview for both MIDUS Core and MIDUS Refresher samples. Participants were asked how many servings and how often they consumed a specific food group during an average day or week. Typical serving sizes for different food types were pre-specified in the food intake questionnaire. Further details on the methodology, including the description of the questions and possible responses, can be found in the public portal at https://midus.colectica.org/ (accessed on 21 September 2024).

### 2.3. MIDUS Healthy Eating Index (MIDUS-HEI)

Food item inclusion and scoring decisions for designing the MIDUS-HEI were made a priori, according to dietary information available in MIDUS and based on previously published HEI, AHEI, and Mediterranean diet-based indices that had shown a consistent association with the risk for chronic disease [14,15,16]. MIDUS-HEI (Table 1) includes 10 food group items: (a) five components (healthy foods such as fruits and vegetables, whole grains, oily fish, lean meat, and non-meat proteins) that were positively related to the index score, (b) three components (unhealthy foods, including sugared beverages, high-fat meat, and fast food) that inversely related to the score, and (c) two components (i.e., fermented dairy and alcohol), which have a non-linear relationship with the total score. Each component contributes 1 point if the participant meets the recommended criteria, 0.5 if she/he meets them partially, and zero if she/he does not meet a healthy intake criterion. Only the consumption of fruits and vegetables contributes 2 points to the total score, because the intake of these two food groups was asked together in a single question. Thus, the MIDUS-HEI score ranged from a low of 0 (very unhealthy diet) to maximal total of 11 points (very healthy diet).

### 2.4. Sociodemographic and Health Information

Sociodemographic variables included age, biological sex, race, and educational level (high school or less, college, and postgraduate studies). Race was ascertained by self-identification. Health-related variables included body mass index (BMI, kg/m^2^), waist circumference (cm), smoking status (never, former, or current), low-density lipoprotein cholesterol (LDL-c, mg/dL), high-density lipoprotein cholesterol (HDL-c, mg/dL), triglycerides (mg/dL), total cholesterol (mg/dL), plasma glucose (mg/dL) and insulin (uIU/mL), HbA1c (%), and HOMA-IR.

BMI and waist circumference were assessed by trained clinical personnel during the participants’ overnight visits. Information about prescription drug usage was also acquired during the visit. All biochemical measurements were derived from a fasting blood sample, as described elsewhere [17], and analyzed by CLIA-certified clinical laboratories. Lipids and HbA1c were measured at the UnityPoint Health-Meriter Laboratory (Madison, WI, USA); glucose and insulin levels were determined by the ARUP Laboratories (Salt Lake City, UT, USA).

### 2.5. Metabolic Syndrome (MetS)

MetS was diagnosed according to the Adult Treatment Panel III (ATPIII) updated definition [18], by the presence of at least three abnormal findings out of the following 5 criteria: (1) central adiposity (waist circumference >102 cm for men and >88 cm for women), (2) elevated triglycerides (≥150 mg/dL) or presence of pharmacological treatment for hypertriglyceridemia, (3) reduced high-density lipoprotein cholesterol (HDL-cholesterol) (<40 mg/dL for men and <50 mg/dL for women) or use of prescription medications for low HDL-cholesterol, (4) elevated blood pressure (systolic ≥130 mmHg or diastolic ≥85 mmHg) or use of hypertensive medications with a prior diagnosis of this condition, and (5) elevated fasting glucose levels (glucose ≥100 mg/dL) or pharmacological treatment for hyperglycemia.

### 2.6. Lipid Profiling

Sera were stored in ultracold freezers and shipped frozen in a single batch to Metabolon, Inc. (Durham, NC, USA) for lipid profiling as part of an untargeted lipidomic analysis (Complex Lipid Panel). Lipid species were quantitatively profiled using LC-MS methods as described elsewhere [19]. Quantitation was achieved using class-specific internal standards, with each lipid class having one or more labeled internal standards. The signal intensity ratio of the target compound to its internal standard was multiplied by the added internal standard concentration to quantify individual lipid species. We included only lipid species for which less than 20% of the data points were below the lower limit of detection to reduced bias and imprecision in data analysis. Lipidomic data included 2072 participants with samples acquired during MIDUS 2 (*n* = 1220) and MIDUS Refresher (*n* = 852).

### 2.7. Lipid Signature of MIDUS-HEI

All lipids were standardized to the same scale using Blom’s transformation [20]. Then, the least absolute shrinkage and selection operator (LASSO) [21] linear regression model was used to regress the MIDUS-HEI on lipid features, allowing one to automatically screen the index-associated features and minimizing the risk of overfitting using the “glmnet” R package [22].

Data from MIDUS Core were used as the training set, and MIDUS Refresher data were considered as the testing set. After training, the lipid signature for the testing set was calculated as the weighted sum of the selected lipids with weights equal to coefficients from the LASSO regression. To avoid training bias, the signature in the training set was obtained using a leave-one-out cross-validation approach. The correlation between the lipid signature and MIDUS-HEI was assessed using Pearson’s r test.

### 2.8. Cardiovascular Risk

A subsample was selected from the MIDUS Core and MIDUS Refresher participants according to the requirements of the 2013 ACC/AHA cardiovascular risk calculation formula. Specifically, all individuals aged 40 to 79 years with total cholesterol levels between 130 and 320 mg/dL, HDL-c between 20 and 100 mg/dL, and systolic blood pressure between 90 and 200 mmHg were included. The resulting subsample included a total of 1450 individuals. To calculate the cardiovascular risk for the selected sample, the 2013 ACC/AHA formula [23] available in UpToDate [24] was extracted and implemented in Python. The estimated cardiovascular risk values were normalized using the Inverse Cumulative Distribution Function, with the stats.norm function from Python’s SciPy library.

### 2.9. Statistical Analyses

The Chi-squared test was used to compare the specific MIDUS-HEI food component score distribution between both MIDUS samples. To examine the associations between MIDUS-HEI score, sociodemographic variables, and nutritional status, a multivariate linear regression was performed, with the following independent variables: time of sampling (MIDUS Refresher sample as reference), sex (women as reference), race (non-white as reference), age (continuous variable, years-old), educational level (postgraduate studies as reference), nutritional status (BMI, continuous variable), and smoking status (current smoker as reference). To analyze the relationship between diet quality, measured by MIDUS-HEI, with metabolic syndrome and its components, logistic regression models adjusted for time of sampling, sex, race, age, educational level, smoking status, and physical activity were used.

To assess the association between the lipidomic signature and cardiometabolic biomarkers, three linear regressions were performed. Model I included adjustment for sample origin and sociodemographic variables (age, sex, and race). Model II included adjustment for both sociodemographic variables and BMI (kg/m^2^), and Model III included an additional adjustment for the MIDUS-HEI score. To evaluate the association between the lipid signature and cardiovascular risk, the same three linear regression models were applied, using the normalized value of the estimated CVR as the dependent variable. For that, the subsample that meets the criteria of the ACC/AHA risk calculator was used. All *p* values were two-tailed and considered to attain statistical significance when *p* ≤ 0.05. The analyses were conducted using SPSS^®^ Statistics (version 24.0), RStudio (version 4.0.3), and Python (version 3.12).

## 3. Results

### 3.1. Sociodemographic Characterization of the Sample and Its Diet Quality

Overall features of MIDUS samples are shown in Table 2. Both samples (Core and Refresher) were similarly represented by sex and were mostly white and middle-aged on average, although ranging from 35–90 years of age, and included representation across multiple levels of educational attainment. Regarding nutritional health, over 75% of individuals were overweight or obese in both samples, and more than 30% met diagnostic criteria for MetS.

Using the algorithm described in Table 1, a MIDUS-HEI score was calculated for each individual in both samples. While the MIDUS Core sample had an average score of 5.59 ± 1.55 points, the MIDUS Refresher sample showed a slightly higher mean score of 5.99 ± 1.52. More detailed information on the consumption of MIDUS-HEI components in both samples can be found in Appendix A. The foods that were reported to have the healthiest consumption in both samples were lean meats, non-meat protein foods, and sugared beverages. Whole grains were consumed at a higher rate by the MIDUS Refresher participants, as compared to the MIDUS Core subsample, perhaps reflecting temporal changes in awareness about the benefits of diet. Conversely, the lowest diet contributors to the HEI score were oily fish, fast food, and alcoholic beverages.

To characterize the relationship between MIDUS-HEI and nutritional and sociodemographic variables, a multivariate linear regression analysis was conducted, including sex, age, race, sample origin, educational level, smoking status, and BMI. As shown in Table 3, all variables demonstrated a significant association with MIDUS-HEI in the multivariate model. Better diet quality was observed in women, white adults, and non-smokers. Additionally, diet quality was positively associated with higher educational level and older age. Regarding nutritional status, a higher BMI was associated with poorer diet quality.

### 3.2. Association of MIDUS-HEI Score with MetS

To analyze the relationship between the MIDUS-HEI score and diet-related chronic diseases, we evaluated whether the index was associated with exceeding the criteria for MetS. As shown in Figure 1, the MIDUS-HEI score was inversely correlated with the prevalence of MetS in the combined data from MIDUS Core and MIDUS Refresher participants (*p*-value for trend: 0.013). Moreover, binary logistic regression models showed that individuals with lower MIDUS-HEI scores were more likely to have abdominal obesity, low HDL-cholesterol, and high fasting glucose, as well as meet criteria for MetS (*p* = 0.012) after adjusting for control variables (e.g., sampling time, sex, race, age, education, and smoking status) (Appendix A). Hypertension and high triglycerides did not show a significant association with the MIDUS-HEI score.

### 3.3. Lipidomic Signature of MIDUS-HEI

Based on the inverse association between MIDUS-HEI and signs of metabolic dysregulation, we proceeded to identify a lipid signature associated with it, by applying machine learning to the lipidomic data. Using a LASSO regression, with MIDUS Core as a training set and MIDUS Refresher as the testing set, we generated a model with 57 lipid species that best predicted MIDUS-HEI. These species were primarily triglycerides (23%), phosphatidylcholines (19%), and ether-linked phosphatidylethanolamines (11%) (Figure 2a). The training set derived from the MIDUS Core participants indicated that lipidomic signature was significantly correlated with MIDUS HEI (r = 0.44, *p* < 10^−5^). The testing set based on the MIDUS Refresher participants verified that the lipidomic signature was significantly correlated with MIDUS-HEI (r = 0.31, *p* < 10^−5^). Figure 2b shows the association between the lipid signature and the MIDUS-HEI scores in both combined sets (r = 0.39, *p* < 10⁻¹⁶).

When evaluating the species with the strongest positive contribution to the MIDUS-HEI signature, the triglycerides and phospholipids rich in polyunsaturated fatty acids stood out. In particular, the fatty acid among these lipids that was most salient, and characteristic of a healthy diet, was docosahexaenoic acid (C22:6). On the other hand, when evaluating those species with the largest negative contribution, triglycerides and phospholipids rich in saturated fatty acids and phospholipids with adrenic acid (C22:4) were prominent (Figure 2c).

### 3.4. Association Between Lipidomic Signature of MIDUS-HEI and Cardiometabolic Biomarkers

To evaluate whether the lipid profile representative of a healthy diet was associated with better cardiometabolic health, multivariate linear regressions were conducted between the lipid signature and several biomarkers indicative of health. After adjusting for sociodemographic variables and BMI, the lipidomic signature associated with MIDUS-HEI showed a significant positive association with HDL-c levels and a significant negative association with insulin levels, HOMA-IR, and the two inflammatory markers (IL-6 and CRP). Interestingly, all these associations remained significant even after adjusting for the MIDUS-HEI score (Table 4).

To evaluate the possible association with CVR, a subsample of the participants meeting the criteria of the ACC/AHA risk calculator was selected. The characteristics of this subsample are shown in Appendix A. As shown in Figure 3, the lipidomic signature of MIDUS-HEI was significantly associated with a lower estimated cardiovascular risk of 9.5% in the model, after adjusting for sociodemographic variables and BMI. Importantly, the magnitude and significance of this association remained similar even after adjusting for the variance accounted for by the MIDUS-HEI score.

## 4. Discussion

This study refined a Healthy Diet Index as a practical tool for assessing dietary quality in a cohort of middle-aged and older American adults and identified its lipidomic signature, demonstrating its association with improved cardiometabolic health.

First, we created the MIDUS-HEI utilizing available food intake data, based on previously validated dietary pattern indices, emphasizing a Mediterranean-style dietary pattern, which has been shown to offer the greatest cardiometabolic benefits [14,15,16]. Our findings indicated that the MIDUS-HEI score is a robust indicator of dietary quality, showing a significant inverse association with unhealthy diet-related biomarkers, adverse health indicators, and MetS—a known precursor to atherosclerotic cardiovascular disease and diabetes.

Then, our aim was to identify the blood lipidomic profile associated with the MIDUS-HEI and evaluate its relationship with cardiometabolic biomarkers and estimated CVR. We identified a lipidomic signature comprising 57 distinct lipid species that was highly correlated with the MIDUS-HEI score. This profile was characterized by lower levels of saturated fatty acids and higher levels of PUFAs. Adults consuming a healthier diet had higher levels of omega-3 PUFAs, precursors of anti-inflammatory molecules, whereas individuals consuming an unhealthy diet had higher circulating levels of some omega-6 PUFAs, precursors of pro-inflammatory eicosanoids. This MIDUS-HEI lipidomic signature was predictive of better cardiometabolic health and reduced cardiovascular risk.

To date, over 2090 studies have utilized MIDUS datasets. Some have treated dietary factors as mediator/moderator variables [25] or as single or composite outcomes [26]. However, the lack of a comprehensive and consistent index for evaluating dietary quality has led to significant variation in how diet is examined and incorporated across different reports. For instance, some investigators considered fruit and vegetable consumption alone as indicators of a healthy diet [27], while others have included other food groups to define diet quality [28]. Notably, some other reports that focused on health behavior and metabolic disease among the participants in the MIDUS project did not include diet as a control variable [29]. As a new alternative option, the MIDUS-HEI score offers a reliable and coherent measure of overall diet quality among MIDUS participants, exhibiting an inverse cross-sectional association with unhealthy diet-related biomarkers and MetS. Our findings align with numerous previous studies that have demonstrated a significant association between a healthier diet and better metabolic health [30]. Furthermore, poor diet quality was significantly associated with the likelihood of exhibiting each component of MetS, except for hypertension and high triglyceride levels. These findings align with extensive literature linking abdominal adiposity with cardiovascular disease and type 2 diabetes mellitus, as well as showing that increased inflammatory activity contributes to onset and progression of diet-related chronic diseases [31]. These associations highlight important policy considerations for promoting population health and the potential for dietary or lifestyle interventions to reduce the prevalence of these disorders.

Dietary surveys are valuable tools for assessing and categorizing diet quality, particularly in population health research. However, they often have several limitations that cannot meet the rigorous standards required for research on nutrition and metabolism, such as the underestimation or overestimation of intake, variation in diet over time, and the complexity of accurately quantifying the nutritional content of foods [10]. Moreover, the accuracy of dietary survey results may be compromised by individual variation in intestinal absorption and metabolism. In this context, generating the lipidomic signature in circulation from blood samples may provide a valuable method for more objectively and comprehensively assessing metabolic responses to healthy and unhealthy diets. In our study, the lipidomic signature showed a significant association with a better lipoprotein profile, improved glucose metabolism, lower levels of pro-inflammatory cytokines, and reduced CVR, even after adjusting for the MIDUS-HEI score. This suggests that the lipidomic signature associated with a healthy diet may be a more sensitive predictor of cardiometabolic risk than the dietary index score alone. The superiority of the lipidomic signature in predicting health outcomes may stem from the fact that it more objectively reflects diet and the person’s metabolic response to the consumed food items and specific nutrients. This finding highlights the complexity of the relationship between lipidomic profiles and cardiometabolic biomarkers, indicating that while diet quality, as assessed by the MIDUS-HEI, plays an important role, other factors may also contribute to the observed lipidomic signature.

An example is adrenic acid; its presence in various lipid classes was consistently associated with an unhealthy diet (i.e., a lower MIDUS-HEI score). Adrenic acid is a fatty acid that is normally found in the human body in small quantities. It can be ingested in small amounts by consuming certain foods, including animal fats and some vegetable oils. However, it is not one of the most common fatty acids in diet, and the primary source of adrenic acid is often endogenous synthesis from arachidonic acid. Arachidonic acid is ingested primarily through animal-based foods such as red meats, poultry, eggs, fish, and dairy products, and it is then converted to adrenic acid mainly through the action of 12-lipoxygenase in various tissues [32]. This conversion can be influenced by other metabolic conditions such as inflammation and oxidative stress. Although evidence suggests that adrenic acid serves as a precursor for both pro-inflammatory and anti-inflammatory eicosanoids, its accumulation appears to enhance inflammation and oxidative stress in a pro-inflammatory setting [33]. For example, in vitro studies and murine models have demonstrated that hepatic accumulation of adrenic acid, in response to increased endogenous synthesis and reduced peroxisomal degradation in the context of steatohepatitis, enhances inflammation and oxidative stress, exacerbating the progression of the pathology [33,34]. Considering the relationship between adrenic acid and an unhealthy diet, as well as its context-dependent effects, provides an example of a lipid that could serve as a sensitive target for more precise nutritional interventions in at-risk populations. In fact, recent Mendelian randomization results suggested that individuals with genetically higher serum adrenic acid levels had an increased risk of coronary heart disease (CHD) events [35], opening new hypotheses about the role of this lipid species.

On the other hand, among the lipid species characteristic of a healthy diet, DHA stood out. Unlike adrenic acid, the primary source of circulating DHA is dietary intake, particularly from the consumption of fatty fish and fish oils. Although the body can synthesize DHA from α-linolenic acid, the efficiency of this conversion is low, making direct dietary intake essential for maintaining adequate circulating levels [36]. Therefore, elevated DHA levels may reflect a higher consumption of these food sources. DHA, the dominant omega-3 in the brain, influences neurotransmitter activity and cognitive functioning. It also plays an important role in promoting antioxidant processes, and a deficiency of DHA has been associated with various neurodegenerative diseases [37]. Considering this, our lipidomic signature of a healthy diet may explain not only the relationship between diet and improved cardiometabolic health but also diet and better mental health.

Notwithstanding the novelty of the current results and the many strengths of a large cohort study, it is important to acknowledge some of the limitations. First, a limitation of our MIDUS-HEI is that, as it is based on the Mediterranean dietary pattern, it incorporates moderate wine consumption with meals. It is important to mention that recent studies suggest caution, particularly regarding the current scrutiny of wine’s role in cardiovascular prevention [38]. Mediterranean indices are adaptable, excluding the alcohol component in specific populations, such as children or individuals with strict alcohol restrictions. In our analysis, however, we have retained the full index to maintain consistency with other studies [12]. Notably, the alcohol component constitutes only a minor fraction of the overall score, and given our primary focus on lipidomic characterization, the single point awarded for moderate alcohol intake has negligible influence on the lipid profile examined.

Another important limitation is that a cross-sectional design precludes definitively drawing conclusions about the causal nature of diet quality and identifying specific mediators and metabolites that initiate pathophysiology and disease outcomes. An assessment of diet at a single time point also does not capture the likelihood of some dietary changes over time. The latter limitation was partially offset by considering two subsamples of the MIDUS cohort who were recruited at different times. In addition, the MIDUS study is ongoing, and the repeated assessments will allow for the evaluation of long-term impact of diet on morbidity. In this context, the MIDUS-HEI and its associated lipidomic signature provide valuable tools for obtaining a more comprehensive understanding of the consequences of eating a healthy or unhealthy diet, as well as other factors that are often antecedent to decisions people make about dietary choices. This is particularly important, given that MIDUS provides a rich platform for investigating the role of various factors (e.g., stress exposures, psychosocial risk and protective factors, and health behaviors) in understanding the relationships between diet, lipidomics, biomarkers, and disease.

## 5. Conclusions

In conclusion, this study successfully refined the MIDUS-HEI, a practical tool for assessing dietary quality. Our findings show that dietary quality scored by MIDUS-HEI is significantly inversely associated with several adverse health indicators and MetS. Furthermore, we identified a lipidomic signature associated with the MIDUS-HEI that was characterized by a favorable balance of specific lipid species and associated with improved cardiometabolic health. Lipidomic profiling offers a promising approach for more objectively evaluating the metabolic consequences of diet and the physiological pathways impacted by healthy and unhealthy diets. Moreover, the analysis identified some lipid metabolites underlying the relationship between dietary behavior and the development of cardiovascular diseases. An awareness of the associations can guide new strategies to more precisely target nutritional interventions for at-risk populations.

## Figures and Tables

**Figure 1 nutrients-16-03995-f001:**
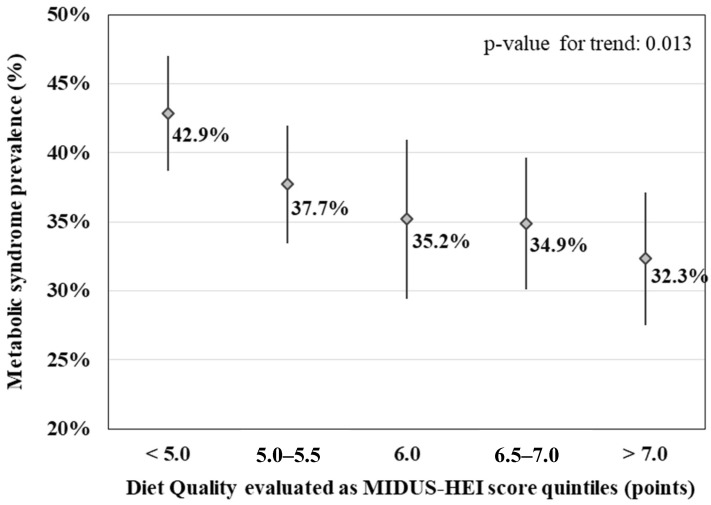
Prevalence of MetS based on diet quality. Diamonds represent the prevalence estimates, and the lines indicate the 95% confidence interval of each estimate.

**Figure 2 nutrients-16-03995-f002:**
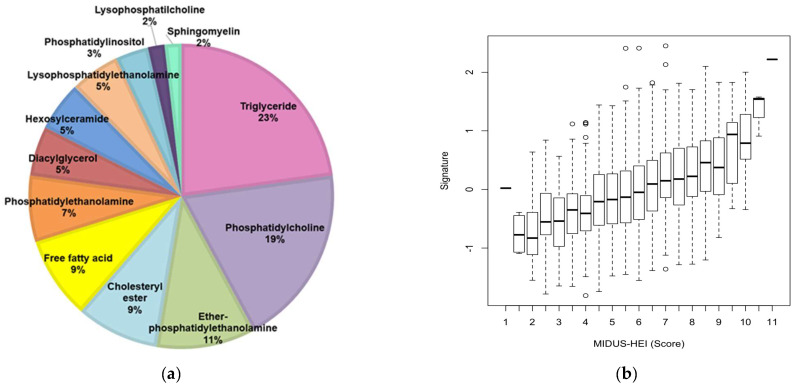
Lipid signature associated with the MIDUS-HEI. (**a**) Percentage of species selected by the lipid signature according to lipid class; (**b**) Association between the score of MIDUS-HEI and lipid signature in the training set and the testing set combined (r = 0.39, *p* < 10⁻^16^); (**c**) Signature coefficients of the 57 metabolites comprising the lipid signature.

**Figure 3 nutrients-16-03995-f003:**
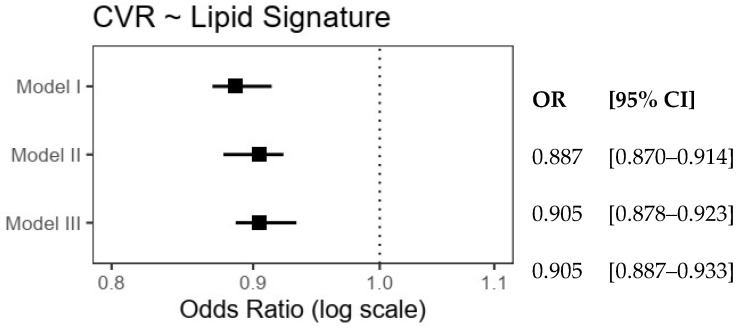
Odds ratios for the association between estimated cardiovascular risk and MIDUS-HEI lipidomic signature. Model I included sociodemographic adjustments, Model II added adjustments for body mass index, and Model III was adjusted further for the MIDUS-HEI score.

**Table 1 nutrients-16-03995-t001:** Scoring of MIDUS Healthy Eating Index (MIDUS-HEI).

MIDUS-HEI Component	Minimal Score (0)	Intermediate Score (0.5 Point)	Maximal Score (1 Point)	Maximal Double Score (2 Points) *
**Direct relation between score and food intake**				
Vegetables and fruits (servings/day)	None	1–2	3–4	≥5
Whole grains (servings/day)	None	1–2	≥3	
Oily fish (servings/week)	None	<1	≥1	
Lean meat (servings/week)	None	0–2	≥3	
Non-meat protein food (servings/week)	<1	1–2	≥3	
**Inverse relation between score and food intake**				
Sugared beverages (servings/day)	≥4	1–3	None	
High-fat meat (servings/week)	≥3	1–2	<1	
Fast food (times/week)	≥1	<1	None	
**Non-linear relation between score and food intake**				
Fermented dairy (servings/day)	<1 or ≥5	(1 to <2) or (4 to <5)	≥2 and <4	
Alcohol (Frequency and quantity)	Nondrinker or (Quantity: Men: >2 drinks/day and Women: >1 drinks/day)	(Frequency: <3 days/week) and (Quantity: Men: 1–2 drinks/day and Women: 1 drinks/day)	(Frequency: ≥3 days/week) and (Quantity: Men: 1–2 drinks/day and Women: 1 drinks/day)	
Total MIDUS-HEI score	0		11	

The bolded titles define the relationship between consumption and the assigned score. * For the combined vegetable and fruit items, the maximal score was 2 points.

**Table 2 nutrients-16-03995-t002:** Sociodemographic characteristics, nutritional status, and metabolic syndrome prevalence by time of sampling in the MIDUS study.

Characteristics	MIDUS Core 59.3% (*n* = 1255)	MIDUS Refresher 40.7% (*n* = 862)
Sex		
Men	43.2 (542)	47.9 (413)
Women	56.8 (713)	52.1 (450)
Age group		
<50 years	29.1 (365)	45.3 (391)
50–65 years	44.8 (562)	38.9 (336)
>65 years	26.1 (328)	15.8 (136)
Race		
White	78.7 (985)	70.2 (606)
Non-white	21.3 (266)	29.8 (257)
Educational level		
Higher education or less	28.0 (350)	17.3 (149)
College education	50.2 (629)	55.0 (474)
Postgraduate studies	21.8 (273)	27.7 (239)
Nutritional status		
Under or normal weight	23.8 (298)	24.8 (214)
Overweight	35.1 (440)	30.0 (259)
Obesity	41.1 (516)	45.2 (390)
Metabolic syndrome		
No	60.8 (750)	65.3 (554)
Yes	39.2 (484)	34.7 (295)

**Table 3 nutrients-16-03995-t003:** Linear regression model between MIDUS-HEI and sociodemographic and nutritional status.

MIDUS-HEI~	Coef (95% CI)	*p*-Value
Sample		
MIDUS Core vs. MIDUS Refresher	−0.467 (−0.612 to −0.323)	0.000
Sex		
Men vs. women	−0.420 (−0.557 to −0.284)	0.000
Race		
White vs. non-white	0.369 (0.207 to 0.531)	0.000
Age		
Age (years old)	0.020 (0.014 to 0.025)	<0.001
Educational level		
Higher education or less vs. postgraduate studies	−0.693 (−0.899 to −0.486)	0.000
College education vs. postgraduate studies	−0.313 (−0.483 to −0.144)	0.000
Nutritional status		
BMI (kg/m^2^)	−0.369 (−0.207 to −0.531)	0.000
Smoking status		
Never vs. current smoker	0.520 (0.310 to 0.731)	<0.001
Former vs. current smoker	0.541 (0.322 to 0.760)	<0.001

The regression coefficients (95% CI) and *p*-values for each variable associated with MIDUS-HEI in the multivariate linear regression analysis are shown.

**Table 4 nutrients-16-03995-t004:** Associations between lipid signature of MIDUS-HEI and cardiometabolic biomarkers.

Variables	Model I ^1^		Model II ^2^		Model III ^3^	
	**Coef**	***p*-Value**	**Coef**	***p*-Value**	**Coef**	***p*-Value**
BMI (kg/m^2^)	−0.213	<0.0001				
LDL-c (mg/dL)	−0.033	0.167·	−0.021	0.376	−0.013	0.614
HDL-c (mg/dL)	0.291	<0.0001	0.227	<0.0001	0.213	<0.0001
HOMA-IR	−0.242	<0.0001	−0.132	<0.0001	−0.119	<0.0001
Insulin (mg/dL)	−0.245	<0.0001	−0.134	<0.0001	−0.120	<0.0001
Glucose (mg/dL)	−0.098	<0.0001	−0.051	0.028	−0.051	0.038
HbA1c (%)	−0.077	<0.001	−0.031	0.169	−0.031	0.186
IL-6 (pg/mL)	−0.263	<0.0001	−0.191	<0.0001	−0.179	<0.0001
CRP (mg/L)	−0.237	<0.0001	−0.142	<0.0001	−0.125	<0.0001

^1^ Model I: Adjusted for sample, age, sex, and race. ^2^ Model II: Model I + BMI. ^3^ Model III: Model II + MIDUS-HEI.

## Data Availability

MIDUS data can be found at https://midus.colectica.org/ (accessed on 21 September 2024). Lipidomic data were deposited to the OSF database under the https://doi.org/10.17605/OSF.IO/VFR7B.

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
