# Peer review of "Lipidomic Signature of Healthy Diet Adherence and Its Association with Cardiometabolic Risk in American Adults"

_nutrients, 2024, doi:10.3390/nu16233995_

Round 1

Reviewer 1 Report

Comments and Suggestions for Authors

This is an interesting cross-sectional study on 2114 adult participants in the “Midlife in the United States study” (MIDUS) aimed to investigate the association between the Healthy Eating Index with metabolic syndrome, lipid signature, cardiometabolic biomarkers and cardiovascular risk. The presentation of results is clear and well described. The discussion and conclusion are consistent with the results obtained. 

Author Response

This is an interesting cross-sectional study on 2114 adult participants in the “Midlife in the United States study” (MIDUS) aimed to investigate the association between the Healthy Eating Index with metabolic syndrome, lipid signature, cardiometabolic biomarkers and cardiovascular risk. The presentation of results is clear and well described. The discussion and conclusion are consistent with the results obtained. 

Response: Thank you for the reviewer’s positive feedback. We hope that the revised version of the manuscript meets all requirements for publication in Nutrients.

Reviewer 2 Report

Comments and Suggestions for Authors

Review of "Lipidomic Signature of Healthy Diet Adherence and Its Association with Cardiometabolic Risk in American Adults " (nutrients-3285692).

This article investigated the utility of Healthy Eating Index (MIDUS-HEI) for diet quality evaluation and the associated blood lipidomic profile to determine its relationship with cardiovascular risk (CVR). Study concepts are interesting; however, this reviewer has several questions and comments.

1.     The primary aim of this study is “refine a Healthy Eating Index (MIDUS-HEI) to evaluate diet quality and assess its association with metabolic disorders”. However, the title emphasizes the second aim. This gives a false impression to the reader.

2.     Methods: 2.2. Dietary Assessment. “Food intake data were obtained via medical history interview conducted by project staff during the biomarker clinic visit.” How exactly were the dietary details interviewed?

3.     2.3. MIDUS Healthy Eating Index (MIDUS-HEI). This reviewer guesses that previous studies have shown  Healthy Eating Index”. It is not clear what makes this MIDUS-HEI different from the previous HEI.

4.     2.6. Lipid profiling. “For our analysis, we considered only lipid species with less than 20% of values below the lower limit of detection.” Why were only lipid species with values less than 20% below the lower limit of detection considered in this analysis?

5.     Results: Table 4. The results of Model 3 showed that lipid signature was associated with various cardiometabolic biomarkers even after adjustment for the MIDUS-HEI score. These results indicate that the lipid signature is also influenced by factors other than the MIDUS-HEI score.

Author Response

Response to Reviewer (2) Comments

Thank you for your time and constructive feedback on our manuscript, "Lipidomic Signature of Healthy Diet Adherence and Its Association with Cardiometabolic Risk in American Adults" submitted to Nutrients. We have addressed each of your comments point by point in the revised manuscript , which we hope meets your expectations and the journal’s standards. All the changes were highlighted in the re-submitted file. Below, you will find our detailed responses to each of your comments.

Point-by-point response to Comments and Suggestions for Authors

Comment 1: The primary aim of this study is “refine a Healthy Eating Index (MIDUS-HEI) to evaluate diet quality and assess its association with metabolic disorders”. However, the title emphasizes the second aim. This gives a false impression to the reader.

Response: Thank you for noting this point. We appreciate your observation regarding the emphasis in the title. To address this, we have rephrased the objectives to be aligned with the primary aim highlighted in the title, ensuring clarity for the reader and consistency throughout the manuscript. The objective of the revised version is as follows: “The aim of this study was to identify the blood lipidomic profile associated with a healthy eating pattern in a middle-aged US population sample, and to determine its relationship with metabolic disorders and cardiovascular risk (CVR).”

Comment 2: Methods: 2.2. Dietary Assessment. “Food intake data were obtained via medical history interview conducted by project staff during the biomarker clinic visit.” How exactly were the dietary details interviewed?

Response: Thank you for your comment. In response, we have added more detailed information in the revised version of the manuscript to provide greater clarity regarding the dietary assessment. For your information, the interview included computerized questions from the Current Health Practices - Diet and Exercise section included in the assessment for both the MIDUS Core and MIDUS Refresher samples. Participants were subjected to a food frequency questionnaire to report how many servings and how often they consumed specific food groups during an average day or week. Each of the food groups queried is described in the MIDUS-HEI section. Typical serving sizes for different food types were pre-specified in the food intake questionnaire. Further details on the methodology, including the description of the questions and possible responses, can be found in the MIDUS public portal at https://midus.colectica.org/.

Comment 3: 2.3. MIDUS Healthy Eating Index (MIDUS-HEI). This reviewer guesses that previous studies have shown “Healthy Eating Index”. It is not clear what makes this MIDUS-HEI different from the previous HEI.

Response: You are correct that previous studies have used various healthy eating indexes (HEIs) as measures of food intake quality. The MIDUS-HEI, however, is an adapted version of previously validated dietary pattern indexes, incorporating the specific dietary questions available in the MIDUS study. In addition to standard HEIs, the MIDUS-HEI also emphasizes some food items included in a Mediterranean-style dietary pattern, which has been shown to offer the greatest benefits for cardiometabolic health. We have clarified this distinction in the revised manuscript to highlight how the MIDUS-HEI differs from the traditional HEIs by incorporating these specific elements.

Comment 4: 2.6. Lipid profiling. “For our analysis, we considered only lipid species with less than 20% of values below the lower limit of detection.” Why were only lipid species with values less than 20% below the lower limit of detection considered in this analysis?

Response: Indeed, we included only lipid species for which less than 20% of the data points were below the lower limit of detection. This approach was used to ensure that the values incorporated in our analysis were more reliable (reducing bias and imprecision) and not heavily influenced by measurement limitations. We have expanded this explanation in the revised manuscript for further clarity.

Comment 4: Table 4. The results of Model 3 showed that lipid signature was associated with various cardiometabolic biomarkers even after adjustment for the MIDUS-HEI score. These results indicate that the lipid signature is also influenced by factors other than the MIDUS-HEI score.

Response: You are correct that the results of Model 3 suggest that the lipid signature is influenced by factors beyond the MIDUS-HEI score. This finding highlights the complexity of the relationship between lipidomic profiles and cardiometabolic biomarkers, indicating that while diet quality, as assessed by the MIDUS-HEI, plays an important role, other factors may also contribute to the observed lipidomic signature. We have discussed this in greater detail in the revised manuscript to better contextualize our findings.

Reviewer 3 Report

Comments and Suggestions for Authors

This is a first class piece of work. I am confident that it will attract much attention and get many citations. However the paper needs some minor editing.

Line 17, there appears to be a word missing after “predictive value of”

Line 63, the authors need to explain the terms metabolomic and lipidomic assay platforms. The authors forget that many readers are likely to be unfamiliar with these terms.

Table 1, at bottom, it appears that light drinkers of alcohol are considered to have a healthier diet than non-drinkers. This is a very dubious claim and is based on older research studies.

Line 148, correct the spelling of HbA1c

Table III. The meaning of the coefficients is rather confusing. It can be deduced from what is stated in lines 249-253. However, I suggest adding a brief explanation below the table.

Line 285. There is a problem with the superscript in the p value (after r = 0.31).

Line 357, change “cardiovascular risk” to CVR

Lines 429-431, this sentence is confusing (“Indeed …. score”).  

Line 435, probably change “fact that is” to “fact that it”

Comments on the Quality of English Language

Excellent

Author Response

Response to Reviewer (3) Comments:

Thank you very much for your kind feedback. We greatly appreciate your positive comments and are pleased that you found the work valuable. We have already incorporated all of your suggestions, and below you will find a detailed response to each of your comments. In addition, all changes are highlighted in the new document.

Point-by-point response to Comments and Suggestions for Authors

  • Comment 1: Line 17, there appears to be a word missing after “predictive value of”

Response: You are correct that a word was missing in that sentence. The corrected sentence now reads: "Food intake data were used to design the MIDUS-HEI and to evaluate its predictive value by examining its association with health variables." We have made this change in the manuscript.

  • Comment 2: Line 63, the authors need to explain the terms metabolomic and lipidomic assay platforms. The authors forget that many readers are likely to be unfamiliar with these terms.

Response: We appreciate your feedback and understand the importance of clarifying these terms for a broader audience. In response, we have added explanations for "metabolomic" and "lipidomic assay platforms" in the revised manuscript to ensure that readers unfamiliar with these concepts can better understand their relevance to the study.

  • Comment 3: Table 1, at bottom, it appears that light drinkers of alcohol are considered to have a healthier diet than non-drinkers. This is a very dubious claim and is based on older research studies.

Response: We appreciate the insight about recent evidence on alcohol consumption and health, particularly regarding the current scrutiny on wine's role in cardiovascular prevention even when used at moderate dosing. While recent studies suggest caution, the dietary index employed in our study is based on the traditional Mediterranean diet pattern, which includes moderate red wine consumption with meals. Mediterranean dietary indexes are adaptable, excluding the item on alcohol intake in specific populations, such as children or individuals with strict alcohol restrictions, reducing the maximum score. For our analysis, however, we have retained the full index to maintain consistency within other studies. Notably, the alcohol component constitutes only a small fraction of the overall score, and given our primary focus on lipidomic characterization, the single point awarded for moderate alcohol intake has negligible influence on the lipid profile examined. We have addressed this limitation explicitly in the revised discussion section.

  • Comment 4: Line 148, correct the spelling of HbA1c

Response: Thank you for pointing this out. We have corrected the spelling of HbA1c in the revised manuscript.

  • Comment 5: Table III. The meaning of the coefficients is rather confusing. It can be deduced from what is stated in lines 249-253. However, I suggest adding a brief explanation below the table.

Response: Thank you for your suggestion. We agree that adding a brief explanation below Table III would enhance clarity. We have now included a concise description to help readers better understand the meaning of the coefficients.

  • Comment 6: Line 285. There is a problem with the superscript in the p value (after r = 0.31).

Response: Thank you for pointing this out. We have corrected the superscript in the p-value after "r = 0.31" in the revised manuscript.

  • Comment 7: Line 357, change “cardiovascular risk” to CVR

Response: We have made the requested change and replaced "cardiovascular risk" with "CVR".

  • Comment 8: Lines 429-431, this sentence is confusing (“Indeed …. score”).  

Response: Thank you for your feedback. We have paraphrased that sentence to improve clarity and avoid confusion. It can now be found in lines 413-416 of the revised manuscript.

  • Comment 9: Line 435, probably change “fact that is” to “fact that it”

Response: Thank you for pointing this out. We have made the correction and changed "fact that is" to "fact that it" in the revised manuscript.

Response to Comments on the Quality of English Language: The revised version of the manuscript was checked by one of our co-authors who is a native English speaker. We hope that this new version accurately represents our research.